# Validity of Ultra-Short-Term HRV Analysis Using PPG—A Preliminary Study

**DOI:** 10.3390/s22207995

**Published:** 2022-10-20

**Authors:** Aline Taoum, Alexis Bisiaux, Florian Tilquin, Yann Le Guillou, Guy Carrault

**Affiliations:** 1Laboratoire Traitement du Signal et de l’Image (LTSI-Inserm UMR 1099), Université de Rennes 1, 35042 Rennes, France; 2Biosency, 35510 Cesson-Sévigné, France

**Keywords:** heart rate variability (HRV), ultra-short-term HRV analysis, wearable devices

## Abstract

Continuous measurement of heart rate variability (HRV) in the short and ultra-short-term using wearable devices allows monitoring of physiological status and prevention of diseases. This study aims to evaluate the agreement of HRV features between a commercial device (Bora Band, Biosency) measuring photoplethysmography (PPG) and reference electrocardiography (ECG) and to assess the validity of ultra-short-term HRV as a surrogate for short-term HRV features. PPG and ECG recordings were acquired from 5 healthy subjects over 18 nights in total. HRV features include time-domain, frequency-domain, nonlinear, and visibility graph features and are extracted from 5 min 30 s and 1 min 30 s duration PPG recordings. The extracted features are compared with reference features of 5 min 30 s duration ECG recordings using repeated-measures correlation, Bland–Altman plots with 95% limits of agreements, Cliff’s delta, and an equivalence test. Results showed agreement between PPG recordings and ECG reference recordings for 37 out of 48 HRV features in short-term durations. Sixteen of the forty-eight HRV features were valid and retained very strong correlations, negligible to small bias, with statistical equivalence in the ultra-short recordings (1 min 30 s). The current study concludes that the Bora Band provides valid and reliable measurement of HRV features in short and ultra-short duration recordings.

## 1. Introduction

In recent years, especially during the COVID-19 pandemic, there has been a growing trend toward remote online health monitoring using wearable devices. These devices are easy to use, inexpensive, and capable of collecting non-invasive and continuous long-term data, allowing for health status monitoring, diagnosis, and disease prevention [1]. Heart rate (HR) and heart rate variability (HRV) are vitally important in e-health applications as they reflect the activity of the cardiovascular autonomic nervous system, thus, the physiological status of a patient [2,3]. They were found to have prognostic value in several cardiovascular and pulmonary diseases [4,5,6]. One of the leading causes of morbidity and mortality among pulmonary diseases is chronic obstructive pulmonary disease (COPD) [7]. It represents high health and economic burden, especially due to exacerbations [8]. Exacerbations are defined as worsening of the symptoms related to COPD and are often related to cardiac autonomic system dysfunction, resulting in changes in HRV [5,6]. Non-invasive and continuous monitoring of the HRV response in COPD patients is highly beneficial to prevent severe exacerbations.

Two main techniques are often used for HR estimation: conventional Electrocardiography (ECG) and photoplethysmography (PPG) [9]. Although the ECG signal has often been used as a reference for HR estimation, the associated monitoring system is not practical and comfortable for continuous monitoring, because it requires the attachment of multiple electrodes to the patient’s chest [10]. In contrast, PPG is a popular, non-invasive technique that is widely applied to modern wearable devices because of its higher usability and lower cost than ECG [10,11]. However, it is more sensitive to body movements and results in noisy waveforms, which affect the HRV measurement. Indeed, the accuracy of PPG in measuring HRV has been validated under controlled and rest conditions [12]; however, it decreases as motion and exercise levels increase [13,14]. 

Long-term (≥24 h) and short-term (~5 min) HRV analyses have been widely studied and physiologically justified [2]. With the development of new wearable device technologies and the emergence of scientific evidence on the potential benefits of continuous HR monitoring, HR data collection has been simplified, and the demand for ultra-short-term (<5 min) HRV analysis has increased [9]. Ultra-short-term HRV analysis is increasingly proposed as a surrogate for short-term HRV analysis. Most studies that have assessed the reliability and validity of ultra-short-term HRV have shown good agreements for time-domain and frequency-domain features on different recordings’ durations [9,15]. Other studies that included nonlinear HRV features showed an increase in error with decreasing duration [9,16]. Further efforts are needed to evaluate the effect of the duration on HRV analysis.

Wearable devices have provided almost real-time measurements and diagnostic tools for acute and chronic health conditions [1,17]. This paper compares a new medical device called the Bora Band to the reference ECG monitor. The Bora Band (Biosency, Cesson-Sévigné, France) was developed for daily remote monitoring of patients with respiratory insufficiency, with an aim to provide a predictive diagnosis of COPD exacerbations. The nominal recording parameters of the Bora Band are a sampling rate of 25 Hz and recording durations of 1 min 30 s every 10 min to allow continuous recording while preserving battery life and storage. Before proposing predictive algorithms of exacerbations using HRV analysis, it is imperative to test the validity of the HRV analysis from the acquired waveforms with the actual nominal recording parameters on a healthy population. Therefore, in this paper, we propose to evaluate the effect of sampling rate and duration on the estimation of HRV features in the time, frequency, and nonlinear domains. We also report the effect of the two parameters on a new set of features computed from visibility graphs, which appear to have high diagnostic significance [18]. We evaluate sampling rates of 25 Hz (nominal) and 200 Hz (up-sampled signals) and durations of 5 min 30 s (short-term) and 1 min 30 s (ultra-short-term) for PPG recordings. Reference HRV features are calculated from ECG recordings of 5 min 30 s duration.

The main contributions of this paper are the comparison between the new device—Bora Band—and the reference ECG monitor in measuring HRV features and the validity of the HRV features extracted from the Bora Band in ultra-short recordings. 

## 2. Materials and Methods

### 2.1. Data Collection

Five healthy adults (3 males, 2 females, 30 ± 9 years old, BMI 21.2 ± 1.8 kg/m^2^) voluntarily participated in this study. The participants were fully informed of the study’s objectives and provided their consent. Data were collected between April and August 2021. Since PPG signals are sensitive to motion artifacts, data collection was performed during inactive hours. Each participant was instructed to wear an ECG monitoring system and the Bora Band overnight from approximately 11 p.m. to 7 a.m. To compensate for the small number of participants, the measurements were performed on 2 to 4 nights depending on the participants, yielding a total of 18 nights of recordings. 

The Bora Band is a connected bracelet worn on the wrist (Biosency, Cesson-Sévigné, France) that integrates an accelerometer, a gyroscope, and a PPG sensor. It records the PPG amplitude on three channels: green, red, and infrared. It has been set to record at a sampling rate of 25 Hz for 5 min 30 s every 10 min during the wearing period. In addition to the raw recordings, the Bora Band can measure the overall heart rate, oxygen saturation level, respiratory rate, activity, and skin temperature of the participants. The ECG was measured using the Actiwave Cardio monitor, recording on a single channel at a sampling rate of 256 Hz with an accelerometer recording at 32 Hz (CamNtech Ltd., Cambridge, UK). The ECG was recorded continuously during the night and was considered the gold standard for HRV computations. Since the ECG was recorded continuously and PPG was recorded for 5 min 30 s every 10 min, each night, a long ECG recording and approximately 54 PPG recordings were obtained, yielding a total of 972 PPG recordings. The monitors used and their placements are shown in Figure 1. 

### 2.2. Data Processing

The processing algorithm is shown in Figure 2. It starts with evaluating the PPG signal quality and eliminating poor-quality recordings, followed by aligning and trimming the continuous ECG recording in accordance with the good-quality PPG recordings. Then, the PPG and ECG data are filtered, and pulse peaks (P-peaks) and R-peaks are detected to calculate pulse-to-pulse and beat-to-beat intervals, P-P intervals, and R-R intervals, respectively. Then, features are extracted from P-P and R-R intervals as surrogates of HRV features. Finally, we evaluate the agreements of HRV features between PPG and ECG at two levels:Recording duration: two recording durations are tested—the default duration of 5 min 30 s and 1 min 30 s. The first 90 s of each recording was considered to obtain the shorter recordings of 1 min 30 s.Sampling rate: As mentioned earlier, the nominal sampling rate for the Bora Band is 25 Hz. However, the temporal resolution over this sampling rate is low compared to ECG. Therefore, the PPG recordings are resampled to 200 Hz using the fast Fourier transform.

#### 2.2.1. Pre-Processing

We began the pre-processing step by identifying and eliminating poor-quality PPG acquisitions based on quality metrics that were computed on 4 s blocks with 2 s overlap. They consist of identifying whether the bracelet is worn, the acquisition is stable, a pulsatile signal is detected, and no important movement is detected.

Then, a 4 s block is considered invalid if any of the above conditions are not met. A PPG recording is considered too noisy and eliminated from the study if there are more than 10% invalid 4 s blocks. Among the selected good-quality PPG recordings, only the PPG amplitude of the green channel was used in the peaks detection algorithm as it is more robust to the movement noise [14]. The green-channel PPG signal was filtered using a forward–backward second-order Butterworth bandpass filter with a frequency band of (0.5, 4 Hz).

The continuous ECG data were sliced into shorter duration recordings of 5 min 30 s and aligned on the remaining PPG recordings by using their respective time stamps. ECG recordings were filtered by a forward–backward second-order Butterworth bandpass filter with a frequency band of (5, 30 Hz). Then, the second derivative of the Actiwave’s accelerometer signal was computed, and ECG values corresponding to high accelerations were replaced by zero.

#### 2.2.2. R-Peak Detection from the ECG

Since PPG and ECG recordings are distinct, two peak detection algorithms were implemented. R-peaks were detected from the ECG recordings using a modified version of the “adaptive and time-efficient algorithm” proposed by Qin et al. [19]. The algorithm was applied using a sliding window of 4 s with a 3 s overlap to account for amplitude changes during the night. ECG data obtained from each window were normalized by dividing the values by their maximum. Then, R-peaks were identified by searching for local maxima and selecting “true” R-peaks based on adaptive amplitude and time interval thresholds.

#### 2.2.3. P-Peak Detection from the PPG

The waveform of the PPG signal indicates the changes in pulsatile blood flow from which the detection of signal peaks allows the calculation of peak-to-peak (P-P) intervals, which translates to a measure of HR [10]. Pulse peaks were detected by searching for minima and maxima whose relative difference exceeds a threshold based on the interquartile range of the data, as proposed by Navarro et al. [20]. We included in the algorithm an adaptive amplitude threshold and a backward search process to improve the algorithm and account for changes in the amplitude of the PPG signal. The first step was to initialize the algorithm on a 10 s window and a fixed threshold. The amplitude threshold was updated according to the detected extrema in the initialization phase. Then, a search for maxima and minima was performed in the signal based on the updated amplitude threshold. The latter was updated at each detection of extrema. If no maximum was detected for more than 5 s, the algorithm returned to the last detected maximum and divided the threshold by two until extrema were detected.

#### 2.2.4. R-R and P-P Intervals Computation

R-R and P-P intervals were calculated from the R-peaks detected from the ECG and the P-peaks detected from PPG recordings, respectively. They were corrected for outliers (intervals < 600 ms and intervals > 1500 ms) and ectopic beats using linear interpolation. Periods of poor-quality PPG recordings were identified in the R-R and P-P intervals of the ECGs and PPGs and removed. These periods are defined by four or more consecutive invalid 4 s blocks. If three invalid 4 s blocks are separated by one valid block, the entire period is considered invalid. Figure 3 illustrates an example of P-P and R-R intervals computation from PPG and ECG recordings in two cases: PPG recording with and without poor-quality periods. In Figure 3A, the PPG recording does not present any invalid period, the resulted P-P interval is completely maintained, and the RR-interval is corrected from ectopic beat. On the other hand, there is an invalid period in the PPG recording, as shown in Figure 3B (highlighted in grey). This period of poor-quality recordings results in a misidentification of peaks and in erroneous P-P intervals. Thus, it is identified and the corresponding R-R and P-P intervals are eliminated (dotted grey lines). For the sake of simplicity, NN intervals will be used in the following to refer to both P-P and R-R corrected intervals. It is worth noting that there is a temporal shift of 2–3 s between ECG and PPG recordings coming from the delay between the internal clock of the two monitors. This delay does not affect the computation of HRV features.

#### 2.2.5. HRV Features Extraction

**Time-domain features:** These included statistical and geometric features, computed using the Neurokit2 Python toolbox. Statistical features computed directly from NN intervals were mean (MeanNN), median (MedianNN), standard deviation (SDNN), coefficient of variation (CVNN), and interquartile range (IQRNN). Features computed from the differences of the successive NN intervals were the root mean square (RMSSD), standard deviation (SDSD), coefficient of variation (CVSD), and the proportion of successive NN intervals differing by more than 20 ms and 50 ms to the total number of NN intervals (pNN20 and pNN50, respectively) [2]. Geometric and distribution-related features included HRV triangular index (HTI), skewness, and kurtosis [21].

**Frequency-domain features:** Since the NN intervals are unevenly sampled and may have missing data due to removed poor-quality periods, the Lomb–Scargle method was used to estimate the power spectral density [22]. Frequency-domain features included low-frequency power (LF; (0.04, 0.15 Hz)), high-frequency power (HF; (0.15, 0.4 Hz)), normalized low- and high-frequency powers (LFnu and HFnu, respectively), and the LF/HF power ratio. The calculation of the power in the very-low-frequency band requires a recording period of at least 5 min [21], which is applicable to short-duration recordings but not to ultra-short duration recordings. Therefore, it was not calculated among the frequency domain features. In addition, the frequency bands were adjusted according to the respiratory frequency measured by the Bora Band, and new frequency-domain features were computed over the adjusted bands [23]. The high-frequency band was centered on the respiratory frequency (F_R_), as F_R_ ± 0.05 Hz, and the low-frequency band was readjusted to eliminate any overlap with the high-frequency band. Frequency-domain features were computed using a custom Python code inspired by the Python pyHRV toolbox. 

**Nonlinear features:** We computed the approximate entropy (ApEn), sample entropy (SampEn), and the short-range fractal correlations from the detrended fluctuation analysis (DFA-α1) [21]. From the Poincaré plot, SD1 and SD2 ellipse standard deviations, their ratio (SD1/SD2), and the ellipse area (S) were computed, as well as the cardiac sympathetic index (CSI) and cardiac vagal index (CVI), both extracted from SD1 and SD2 [24]. These features were computed using the Neurokit2 Python toolbox. We computed the acceleration (AC) and deceleration (DC) capacities of the heart rate [25], their total contributions (Ca, Cd), and the total variances of their contributions (SDNNa, SDNNd) using a custom Python code following the algorithm proposed by Piskorski et al. [26]. Moreover, visibility indices were computed from the global visibility graph (VG) and the horizontal visibility graph (HVG) using a MATLAB code and the MATLAB networks toolbox [27,28]. They included the mean degree (MD-VG, MD-HVG) of the nodes, the clusters coefficient (C-VG, C-HVG), the transitivity (Tr-VG, Tr-HVG), and the assortativity (r-VG, r-HVG) of the visibility graphs.

In the following, the HRV features extracted from the 5 min 30 s ECG recordings and PPG recordings are defined as HRV_E5_ and HRV^F^_PX_, respectively. E and P denote the ECG and PPG recordings. X represents the duration of recordings; X = 1 if recordings of 1 min 30 s duration are considered, and X = 5 if recordings of 5 min 30 s duration are considered. As previously noted, ECG recordings are analyzed for a duration of 5 min 30 s only and considered as the reference for the comparisons. F is the sampling rate of the analyzed PPG recordings of 25 Hz or 200 Hz.

### 2.3. Statistical Analysis

A Shapiro–Wilk test was performed to assess the normality of the data. As the data were not normally distributed, the extracted features were transformed using the natural logarithmic (log) transformation when necessary to allow for parametric statistical comparisons that assume normality. As previously mentioned, multiple recordings were obtained for each night and for each participant, which increases the number of measurements analyzed but increases the inter-subject variability. The agreement between HRV^F^_PX_ and HRV_E5_ was evaluated using different techniques that were adapted to account for repeated measures and within-subject variability. First, the correlation between HRV_E5_ and HRV^F^_PX_ was tested by calculating a repeated-measures correlation coefficient with 95% confidence intervals to consider the within-subject variability using the Pingouin Python package [29,30]. The correlation was defined as poor if the coefficient was <0.3, fair (<0.6), moderately strong (<0.7), strong (<0.9), and nearly perfect (>0.9).

Although correlation gives an indication of the strength of the relationship between the two methods, it does not necessarily guarantee agreement [9]. Therefore, Bland–Altman plots were constructed with mixed-effects limits of agreement (LoA) [31] to assess changes in bias by modifying duration or frequency. Differences in log-transformed HRV features (log(HRV^F^_PX_) − log(HRV_E5_)) were considered. The Bland–Altman method with mixed-effects LoA measures the mean bias (95% LoA) and the within-subject standard deviation with 95% confidence intervals. The Bland–Altman plots were constructed on RStudio using the code provided by Parker et al. [31].

In addition, the non-parametric Cliff’s delta (δ) was used to test the agreement between the two methods. It was computed using the cliffs-delta Python package. A value of |δ| < 0.11 was considered negligible, |δ| < 0.28 a small effect, |δ| < 0.42 a moderate effect, and |δ| > 0.42 a large effect [32].

Finally, a non-parametric two one-sided equivalence test (TOST) was performed to test for equivalence, rather than differences, as in standard statistical tests [33]. Hence, an equivalence region of ±10% from the mean of the HRV_E5_ features was considered. Then, the null hypothesis of non-equivalence was tested separately on either side of the equivalence region to check if the HRV^F^_PX_ features fell outside this region. The null hypothesis was rejected if both one-sided tests were rejected, indicating statistical equivalence. A *p*-value < 0.05 was considered significant. The TOST method was applied using the TOSTER package on RStudio.

## 3. Results

ECG and PPG measurements were recorded for 18 nights. Recordings from two nights were excluded from the study because the Actiwave’s accelerometer signal was not recorded on one night and the Bora Band was set to record PPG data for durations of 1 min 30 s on the other night. From a total of 936 PPG recordings, 449 recordings with a duration of 5 min 30 s and 429 recordings with a duration of 1 min 30 s were maintained after an initial selection using the aforementioned quality metrics. These measurements were filtered and processed to extract HRV features that were compared with those extracted from ECG recordings of 5 min 30 s duration. The median and interquartile ranges of HRV_E5_ and HRV^F^_PX_ features are provided in Appendix A. Results of the statistical comparisons between HRV_E5_ and HRV^F^_PX_ features and between HRV^F^_P5_ and HRV^F^_P1_ can be found in Appendix A. Bland–Altman plots for a selection of HRV features can be found in Appendix A. 

### 3.1. Comparison between ECG and PPG Measurements

A first comparison was performed between the HRV_E5_ features and the features from the Bora Band with the initial settings of 5 min 30 s duration and 25 Hz sampling frequency. Table 1 presents the HRV features where agreement between PPG and ECG recordings are validated with a correlation coefficient > 0.7 and Cliff’s delta < 0.28. All time-domain features, except for the geometric ones, have high correlations and low Cliff’s delta between PPG and ECG recordings. HRV^25^_P5_ features extracted directly from NN intervals are significantly equivalent to HRV_E5_ features (*p*-value < 0.05); however, those computed from the successive differences of NN intervals (RMSSD, SDSD, CVSD) are not statistically equivalent. All frequency-domain features have high correlations and low Cliff’s delta. LF and LFnu in the standard and adjusted frequency bands and HFnu in the adjusted frequency bands are statistically equivalent between ECG and PPG recordings. For nonlinear features, only those shown in Table 1 have high correlations and low Cliff’s delta, while all other extracted features have very low correlations between ECG and PPG recordings. Although SD1, SD1/SD2, S, and CSI show good agreement between ECG and PPG, they are not statistically equivalent. Among all the features of visibility graphs, only C-VG shows acceptable agreement between ECG and PPG recordings.

### 3.2. Effect of the Duration of the Recordings

Figure 4 illustrates the change in mean bias and 95% LoA for LF (F_R_) and DFA-α1 as the duration of PPG recordings decreases from 5 min 30 s to 1 min 30 s. Detailed results for the HRV features can be found in Appendix A. As the duration of PPG recordings decreases, the absolute value of the mean bias and the 95% LoA increase for almost all the extracted HRV features. In addition, correlation coefficients decrease, and fewer parameters show statistical equivalence between ECG and PPG recordings. HRV^25^_P1_ features that present a good agreement with HRV_E5_ are MeanNN, MedianNN, SDNN, pNN50, HF (F_R_), LFnu (F_R_), SD2, S, CVI, SDNNa, and SDNNd, as shown in Table 2.

It could be contested that the effect of duration is assessed by comparing two different waveforms recorded at different sampling rates. To this end, we evaluated the agreement between short- and ultra-short-term HRV features extracted from PPGs recorded at a sampling rate of 25 Hz. Table 3 shows the HRV features for which the agreement between the 5 min 30 s and 1 min 30 s PPG recordings is validated by a correlation coefficient > 0.7, Cliff’s delta < 0.28, and a validated equivalence test with a *p*-value < 0.05. The detailed results of the statistical tests can be found in Appendix A.

### 3.3. Effect of the Sampling Rate

As previously mentioned, we evaluated the effect of upsampling the PPG waveforms to 200 Hz over the default sampling rate of 25 Hz. Figure 5 shows the change in mean bias and 95% LoA for MeanNN, RMSSD, LFnu, SD1, DFA-α1, and Tr-HVG as the sampling rate of recordings increases for the two durations of 5 min 30 s and 1 min 30 s. As can be seen, increasing the recording frequency decreases the absolute value of the mean bias and the 95% LoA for almost all the extracted HRV features and for both recording durations. This can be explained by the increased temporal resolution of the waveforms (which is eight times higher at 200 Hz than at 25 Hz) and, thus, of the detected NN intervals.

Table 4 lists the HRV^200^_P5_ and HRV^200^_P1_ features, for which the agreement was validated with the reference HRV_E5_ features. Most HRV features in the time domain show good agreement between PPG and ECG for short and ultra-short recordings. However, the agreements for HRV features in frequency-domain, nonlinear, and visibility graphs are different depending on the recording durations. Those with good agreement between HRV^200^_P1_ and HRV_E5_ are: HF and LFnu in the adjusted frequency bands, SD1, SD2, S, and CVI from Poincaré plots, and SDNNa and SDNNd from accelerations and decelerations.

## 4. Discussion

The objective of this study was to assess the validity of the Bora Band PPG recordings for measuring short-term HRV compared with the reference ECG in healthy subjects. The effects of increasing the sampling frequency and decreasing the recording duration (ultra-short duration) were tested against standard 5 min 30 s ECG recordings. Additionally, the validity of ultra-short-term HRV analysis was evaluated by comparing ultra-short and short PPG recordings of the same sampling frequency.

The literature reviews identified studies that addressed the validity and reliability of ultra-short-term HRV analysis compared with short-term recordings [9,15]. Most existing studies have assessed agreement using correlation and/or statistical differences tests. If performed alone, these tests are insufficient to draw conclusions about the reliability of shorter-term recording because they do not control for measurement bias. Of the studies identified, only three studies assessed measurement bias using Bland–Altman plots with LoA followed by Cohen’s d statistic [9]. Since then, recent studies have followed the recommendations of Shafer et al. and Pecchia et al. [9,15] to assess the reliability of ultra-short HRV analysis [34,35,36]. To our knowledge, there are no studies that have evaluated the reliability of ultra-short-term HRV features using equivalence tests; instead, they have used standard statistical tests of difference. Applying these tests is inappropriate since they are intended to detect differences, not equivalence; non-significant group differences are not necessarily evidence of equivalence [9,37]. Therefore, in the current study, we evaluated the agreement between HRV features extracted from PPG and ECG recordings using the correlation coefficients, Cliff’s delta, and Bland–Altman plots, as recommended in the literature [9,15], but also tested the equivalence between the two methods using equivalence tests.

This study showed that the Bora Band is valid for measuring all short-term (5 min 30 s) HRV features in the time and frequency domains and most nonlinear and visibility graphs features. Upsampling the PPG recordings to 200 Hz provided adequate temporal resolution with respect to the reference ECG recordings, resulting in better agreements with reference HRV features. Consistent with the findings of Shaffer et al. [21], this paper showed that the low sampling frequency influenced the validity of the HRV features. Indeed, time-domain features extracted from successive differences of NN intervals, such as RMSSD, SDSD, CVSD, and pNN20, showed lower correlation coefficients, higher Cliff’s delta, and non-equivalence when extracted from PPGs recorded at 25 Hz rather than 200 Hz because these features depend directly on the temporal resolution of the processed data. Similar trends were observed for frequency-domain features, nonlinear features, and visibility graphs features. In this study, the ECG was recorded at a sampling rate of 256 Hz, whereas the PPG was recorded at a sampling rate of 25 Hz, which is 10 times lower than the sampling rate of the ECG. The heart rate signal obtained from the PPG has a temporal resolution (40 ms) that is 10 times less than that obtained from the ECG (4 ms), which induces a difference in the HRV features extracted from the two recordings and affects the agreement and correlation. In addition, the distinct nature of the two waveforms affects the calculation of HRV features and, thus, the agreement. Based on these findings, the Bora Band might be set to record with a sampling frequency of 25 Hz; then, the PPG recordings could be upsampled to 200 Hz in the processing algorithm to increase the temporal resolution. This way, battery life and storage would be preserved while ensuring agreement with ECG reference data.

The validity of the ultra-short-term HRV features was assessed by comparison with the short-term HRV features computed first from the standard goal for HRV features computation (ECG) and then from the same signal (PPG). The results showed concordance between the two validity assessments. In general, research on assessing the validity of ultra-short-term (<2 min) HRV features against short-term features using robust methodological evidence and a large set of HRV features is limited. Due to their practical use, most studies have considered only time-domain features or have added frequency-domain and some nonlinear features. In the time-domain, the main features whose validity was evaluated in ultra-short recordings were the MeanNN, SDNN, RMSSD, and pNN50 [34,35,36,38,39,40], tested alone or in combination with other HRV features. In agreement with our results, these studies found that the tested time-domain HRV features provided strong correlations and high agreement with small bias whenever the latter was assessed.

However, we could not draw a common conclusion from studies that evaluated the validity of ultra-short-term frequency-domain features of HRV [34,35,38,40]. As with our study, the same frequency-domain features were evaluated in ultra-short length ECG recordings using correlation and Bland–Altman plots with a 95% LoA, followed by a Student’s *t*-test [40]. Only LF was valid in 1 min 30 s recordings, whereas HF, LFnu, HFnu, and LF/HF were not valid in recordings shorter than 3 min. LFnu and HFnu also required at least 3 min of recording to be valid with 5 min ECG recordings [34]. Conversely, Wehler et al. tested the validity of ultra-short LF and HF features from ECG to find that both features were reliable and valid on 1 min 30 s recordings based on the assessment of correlations, Cliff’s delta, and log-transformed Bland–Altman plots with 95% LoA [35]. Likewise, strong correlations were found for HF between 1 min and 5 min ECG recordings [38]. In the latter study, LF and LF/HF were not assessed in the 1 min recordings because of theoretical doubts about the loss of information for these features. It is noteworthy that all these studies extracted frequency-domain features over the standard frequency bands and used different methods to estimate the spectral density. According to our results, the HF computed over the standard frequency band was valid only in the 1 min 30 s duration recordings when compared with the 5 min 30 s PPG recordings and not with the 5 min 30 s ECG recordings. However, HF and LFnu, computed over the adjusted frequency bands, provided a surrogate for the 5 min 30 s features computed from both PPG and ECG recordings.

Agreement and validity studies of nonlinear HRV features included SD1 and SD2 from Poincaré plots, ApEn, SampEn, and DFA-α1 [38,40]. These two studies showed that only SD1 and SD2 were valid on recordings ≤ 1 min 30 s. These findings were consistent with those shown in the current study. In addition, in this study, we tested a broader set of nonlinear features to find that most features computed from the Poincaré plots and SDNNa and SDNNd of accelerations and decelerations were valid in ultra-short duration recordings. Moreover, this is the first study that evaluated the features from visibility graphs for PPG-ECG agreement and their validity in ultra-short duration recordings. Features related to connectivity between nodes of global visibility graphs showed good agreements between ECG and PPG for short-term recordings, but none of the features were valid in ultra-short length recordings.

Despite the confidence in the results reported in this paper, this study has some limitations. The results presented in this study were measured from a small number of participants. As only five participants were available for the current study, our results need to be validated on a larger database. Second, the reader should keep in mind that the results are valid for a comparison with an ECG sampled at 256 Hz. It is worth recalling that HRV is often studied with an ECG sampled at 1000 Hz and that a dedicated study should have been carried out at this sampling rate.

## 5. Clinical Interest

The analysis of HRV allows for the profiling of the variability and complexity of the heart rhythm and the autonomic nervous system involved. HRV features were first associated with health to predict fetal distress [41]. Then, many studies were conducted to understand HRV and assess HRV analysis in the prevention of diseases or mortality [2,9,21]. As acute exacerbations of COPD are one of the main causes of morbidity and mortality worldwide and result in high healthcare costs, they need to be detected and treated early. COPD has been associated with functional alterations in sympathetic and parasympathetic activities, which could be studied using HRV features [6]. Most studies that assessed HRV analysis in COPD populations included stable COPD patients or exacerbated COPD patients 24–48 h after treatment [5,6]. A limited number of studies have proposed longitudinal designs for COPD patients based on monthly home visits to record physiological signals [42]. These designs do not provide continuous monitoring of patient health status, making early detection of exacerbations difficult.

As previously mentioned, the ultimate goal of the Bora Band is its implementation in the continuous monitoring of COPD patients in their daily life to detect exacerbations early, before their occurrences. In this way, patients with exacerbation symptoms could be taken care of by clinicians to start the treatment and avoid hospital admission. Future work would address the monitoring of the valid ultra-short-term HRV features in COPD patients and the implementation of an algorithm for predicting exacerbations.

## 6. Conclusions

Overall, this study is the first to provide methodologically complete and robust evidence for the evaluation of the agreement of HRV features between ECG and PPG recordings in short-term recordings and of the reliability of PPG in ultra-short-term recordings. This allowed us to draw certain conclusions about the validity of HRV analysis extracted from the Bora Band recordings over short and ultra-short durations. According to the results, recording PPG at 25 Hz for durations of 5 min 30 s every 10 min would provide the best agreement with ECG reference data if the PPG recordings were upsampled to 200 Hz in the processing algorithm. If there is a need to shorten the recording duration (to spare battery power, for instance), one should be careful when extracting the HRV features because not all of them were valid in the ultra-short recordings. To confirm these findings, it would be interesting to consider a larger database with more participants included in the experiment. Further work would be to evaluate HRV analysis for the prediction of exacerbations in COPD patients.

## Figures and Tables

**Figure 1 sensors-22-07995-f001:**
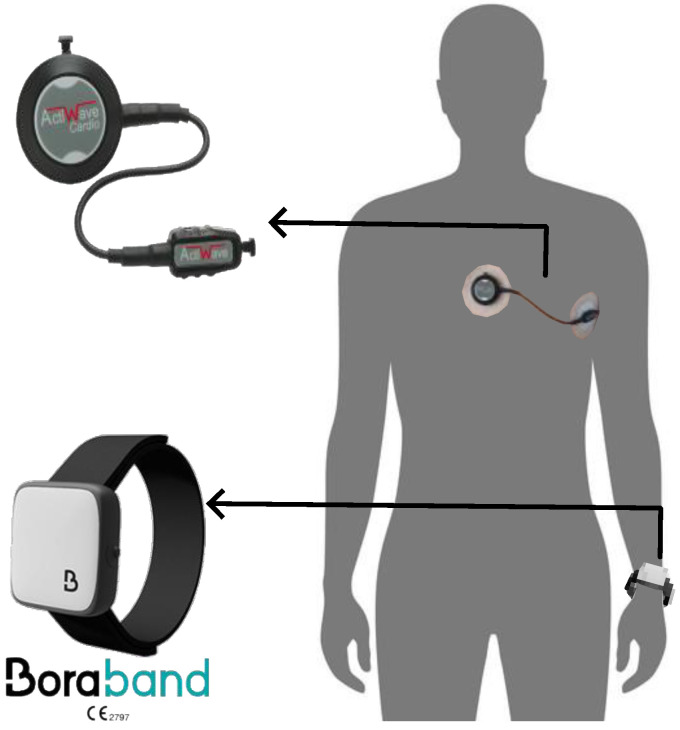
Actiwave and Bora Band placements during acquisitions.

**Figure 2 sensors-22-07995-f002:**
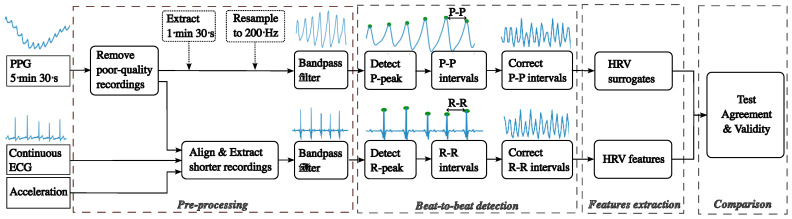
Block diagram of the proposed approach.

**Figure 3 sensors-22-07995-f003:**
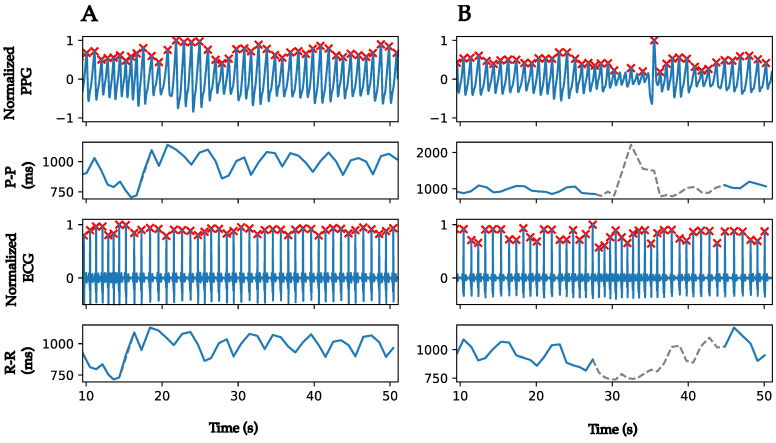
Examples of PPG and ECG recordings with corresponding peaks detection and peak-to-peak intervals. (**A**) Peaks detections from a good-quality PPG recording and an ECG recording. (**B**) Peaks detections and poor-quality periods identification from a noisy PPG recording and an ECG recording. Red cross markers represent the peaks detected. In the P-P and R-R intervals plots, the grey dotted lines present the initial computed P-P and R-R intervals that were eliminated in the correction phase, and the blue plain lines present the NN intervals after correction of outliers, ectopic beats, and removal of poor-quality periods.

**Figure 4 sensors-22-07995-f004:**
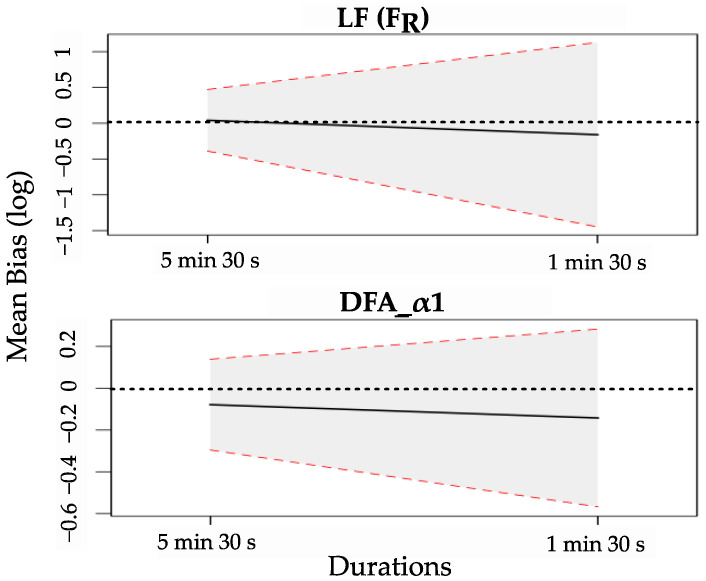
Changes in mean bias and 95% limits of agreement (LoA) for low frequency and DFA-α1 features with decreasing the duration of recording from 5 min 30 s to 1 min 30 s. The black line indicates the mean of observed bias, the dotted red lines show the intervals defined by the 95% LoA, and the dotted black line represents the line of equality with null bias.

**Figure 5 sensors-22-07995-f005:**
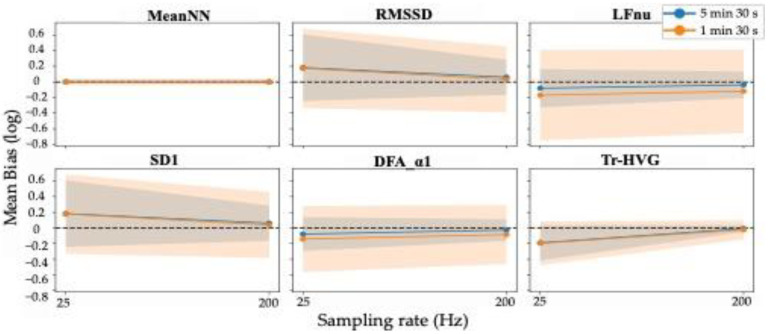
Mean bias changes and 95% LoA of a selection of some HRV features for both durations of 5 min 30 s and 1 min 30 s with increasing the sampling rate from 25 Hz to 200 Hz. Blue and orange lines and shadows indicate the mean of observed bias and the 95% LoA for 5 min 30 s and 1 min 30 s, respectively. The dotted black line is the line of equality with null bias.

**Table 1 sensors-22-07995-t001:** Features showing good agreement between PPG 25 Hz (5 min 30 s) and ECG recordings.

	Correlation Coefficient	Cliff’s Delta (|δ|)	**Equivalence Test *p*-Value**
**Time-domain features**			
MeanNN, MedianNN, SDNN, CVNN, IQRNN, pNN50	>0.7	|δ| < 0.11	<0.05
RMSSD, SDSD, CVSD	>0.7	0.11 < |δ| < 0.28	≥0.05
**Frequency-domain features**			
LF, LF (F_R_), LFnu, LFnu (F_R_), HFnu (F_R_), LF/HF (F_R_)	>0.7	|δ| < 0.28	<0.05
HF, HF (F_R_), HFnu, LF/HF	>0.7	|δ| < 0.28	≥0.05
**Nonlinear features**			
DFA-α1, SD2, CVI, SDNNa, SDNNd	>0.7	|δ| < 0.28	<0.05
SD1, SD1/SD2, S, CSI	>0.7	|δ| < 0.28	≥0.05
**Visibility graph features**			
C-VG	>0.7	|δ| < 0.28	<0.05

**Table 2 sensors-22-07995-t002:** Features showing good agreement between PPG 25 Hz (1 min 30 s) and ECG recordings.

	Correlation Coefficient	Cliff’s Delta (|δ|)	Equivalence Test *p*-Value
**Time-domain features**			
MeanNN, MedianNN, SDNN, pNN50	>0.7	|δ| < 0.11	<0.05
RMSSD, SDSD	>0.7	0.11 < |δ| < 0.28	≥0.05
**Frequency-domain features**			
LFnu (F_R_)	>0.7	|δ| < 0.28	<0.05
HF (F_R_)	>0.7	|δ| < 0.28	≥0.05
**Nonlinear features**			
SD2, S, CVI, SDNNa, SDNNd	>0.7	|δ| < 0.28	<0.05
SD1	>0.7	|δ| < 0.28	≥0.05

**Table 3 sensors-22-07995-t003:** Features showing good agreement between 1 min 30 s and 5 min 30 s PPG recordings at 25 Hz.

	Agreements between HRV^25^_P1_ and HRV^25^_P5_
**Time-domain features**	MeanNN, MedianNN, SDNN, RMSSD, SDSD, CVSD, pNN20, pNN50
**Frequency-domain features**	HF, HF (F_R_), LFnu (F_R_)
**Nonlinear features**	SD1, SD2, SD1/SD2, S, CVI, SDNNa, SDNNd

**Table 4 sensors-22-07995-t004:** HRV features showing good agreement between PPG and ECG recordings. All features have strong correlation coefficients (>0.7), negligible Cliff’s delta (δ < 0.11), and statistical equivalence (*p*-value < 0.05).

Features	5 min 30 s	1 min 30 s
**Time-domain**	MeanNN, MedianNN, SDNN, CVNN, IQRNN, RMSSD, SDSD, CVSD, pNN20, pNN50, kurtosis	MeanNN, MedianNN, SDNN, IQRNN, RMSSD, SDSD, pNN20, pNN50
**Frequency-domain**	LF, LF (F_R_), HF, HF (F_R_), LFnu, LFnu (F_R_), HFnu, HFnu (F_R_), LF/HF, LF/HF (F_R_)	HF (F_R_), LFnu (F_R_)
**Nonlinear**	SampEn, DFA-α1, SD1, SD2, SD1/SD2, S, CSI, CVI, AC, DC, SDNNa, SDNNd	SD1, SD2, S, CVI, SDNNa, SDNNd
**Visibility**	MD-VG, C-VG, Tr-VG, Tr-HVG	-

## Data Availability

This study was conducted by an industrial company (Biosency), and, for protection and confidentiality reasons, the database will not be shared.

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
