# Peer review of "Validity of Ultra-Short-Term HRV Analysis Using PPG—A Preliminary Study"

_sensors, 2022, doi:10.3390/s22207995_

Round 1

Reviewer 1 Report

Results and conclusions are supported by the methods and collected data. However, the study focuses on a very little number of healthy participants.

Summary of participants: gender, age and BMI should be included and results discussed projecting the assessment of short HRV in patients. I also note that recording have been performed at night, I am assuming that none of the participants has sleep disturbances, sleep disturbances can affect HR and HRV. I suggest that "preliminary" is also included in the title.

Paper would benefit of a larger clinical evaluation.

Author Response

We would like to thank the reviewer for the careful and thorough reading of the paper and for the thoughtful comments, which help us improve the quality of the paper. Our detailed responses are presented below.

  • We agree with the reviewer that the BMI should be added to the gender and age presented in Section 2.1. The BMI is added directly in the text in Section 2.1, line 85.
  • We apologize to the reviewer but we did not fully understand their comment “and results discussed projecting the assessment of short HRV in patients”. In fact, the main objective of the company Biosency is to develop a monitoring system (including the Bora Band) for patients suffering from respiratory insufficiency and implement predictive models for exacerbations using HRV features as described in Section 1, lines 65 - 70. As described, the nominal recording parameters of the Bora Band are a sampling rate of 25 Hz and a recording duration of 1 min 30 s every 10 min. Before implementing HRV-based predictive models on COPD patients, it was important to test the validity of the Bora Band with the actual nominal recording parameters on a healthy population, hence the study proposed in this paper. This point is clarified in the paper in Section 1, lines 70-73.
    As mentioned in Section 5 “Clinical interest”, few studies have proposed to monitor COPD patients using HRV features. Our future works would address the issue of evaluating short or ultra-short HRV analysis in the COPD population, as mentioned in Section 5, lines 443 – 448, and Section 6, line 472.

  • We thank the reviewer for their interesting comment. Indeed, sleep disturbance can affect heart rate and heart rate variability. All of the participants included in the study were healthy, and none of them reported major sleep disturbances during the experiment. The objective of this study was to design a proof of concept that the bora band would be interesting to assess heart rate variability by PPG. In this sense, it was natural to start with a healthy population.

  • As noted at the end of the discussion, the study's main limitation is the small number of participants included. Therefore, the results should be considered preliminary and need to be validated on a larger database. On this basis and at the suggestion of the reviewer, the title is changed to “Validity of ultra-short-term HRV analysis using PPG – A preliminary study”.

Reviewer 2 Report

The authors validated ultra-short-term (90s) HRV analysis using PPG by comparing it with short-term (330s) HRV analysis. My main concerns and questions are as follows:

1. Even as a preliminary experiment, five subjects are too few to ensure reliability of the validation. It is necessary to calculate the appropriate number of subjects according to statistical analysis using G-Power even if the effect size is reduced.

2. It is necessary to present the author's physiological interpretation of the cause of uncorrelated features.

3. Why is the VLF(very low frequency) in the frequency domain features not included in the analysis? It is important HRV feature that indicates sympathetic activity.

Author Response

We would like to thank the reviewer for the careful and thorough reading of the paper and for the thoughtful comments, which help us improve the quality of the paper. We apologize if the presentation of the paper was not adequate. We hope that the new version of the paper will provide clarification and satisfaction to the reviewer. Our detailed responses are presented below.

1. Even as a preliminary experiment, five subjects are too few to ensure reliability of the validation. It is necessary to calculate the appropriate number of subjects according to statistical analysis using G-Power even if the effect size is reduced.

We agree with the reviewer that the number of participants is the main limitation of our study. The recruitment was conducted among company employees and included only healthy subjects. Only 5 healthy subjects volunteered for the study. To compensate for the small number of participants, the measurements were performed over several nights (2 to 4 nights depending on the participants). During each night (from ~11 pm to ~7 am), the ECG monitor was continuously recording and the Bora Band was recording for 5 min 30 s every 10 minutes. Recordings from 16 nights were analyzed and a total of 429 recordings of 1 min 30 s duration were considered for the extraction of HRV features. This increased the number of measurements analyzed while increasing the inter-subject variability. Therefore, agreements were assessed using statistical analysis methods that account for repeated measures and within-subject variability. More clarification on the number of recordings was added to the paper in Section 2.1, lines 90-93.

As noted at the end of the discussion, the study's main limitation is the small number of participants included. Therefore, the results should be considered preliminary and need to be validated on a larger database. Based on the above and the suggestion of reviewer 1, the title is changed to “Validity of ultra-short-term HRV analysis using PPG – A preliminary study”.

2. It is necessary to present the author's physiological interpretation of the cause of uncorrelated features.

We thank the reviewer for this comment but we believe that the cause of the uncorrelated features is not necessarily interpreted by physiological aspects. We believe that the difference in HRV features between ECG and PPG is mainly due to the fact that the two waveforms are distinct and measured differently, the different sampling rates of recordings and the difference in the duration of the recordings compared.

The ECG is a direct measure of cardiovascular activity and heart rate. PPG is a measure of blood volume changes in the microvascular tissue bed. The PPG waveform includes the pulsatile physiological waveform that is synchronized with each heartbeat and is therefore considered a mirror of the heart rate. The reliability of the heart rate measured from the PPG depends on various factors, such as the sampling rate of the PPG recordings, motion, etc. While the bracelet measuring PPG waveforms was worn on the wrist, the ECG electrode was attached directly to the chest and is more robust to movement than the PPG. The small movements of the participants that were maintained in the recordings (not eliminated in the preprocessing step) induced errors in the detection of P-peaks and thus the P-P intervals from which HRV features are extracted.

In this study, the ECG was recorded at a sampling rate of 256 Hz, whereas the PPG was recorded at a sampling rate of 25 Hz, which is 10 times lower than the sampling rate of the ECG. The heart rate signal obtained from the PPG has a temporal resolution (40 ms) that is 10 times lower than that obtained from the ECG (4 ms), which induces a difference in the HRV features extracted from the two recordings and affects the agreement and correlation. In this paper, we proposed to resample the PPG recordings to 200 Hz using the Fast Fourier transform (FFT) to increase the temporal resolution of the PPG recordings and HR to 5 ms. This aspect was presented in the discussion in Section 4, lines 374 – 394. It is important to keep in mind the nature of each signal. ECG recordings are known for their sharp shape with the QRS peak, which allows easy and accurate detection of R peaks and calculation of RR intervals. On the contrary, PPG recordings have slower fluctuations than ECG and therefore the detection of P-peaks is less accurate, which affects the calculation of P-P intervals and the extraction of HRV features. Therefore, an error remained between the ECG and the PPG in the extraction of HR and HRV.

Finally, the validity of the ultra-short PPG recordings was assessed by extracting the first 90s from the PPG recordings and comparing the extracted HRV features with those extracted from the 330s ECG recordings. The 90s PPG recordings do not include all the information provided by the 330s PPG recordings. As a result, the error between the ultra-short PPG and the short ECG increased, and the agreement and correlations decreased.

We have added some clarification concerning this point in Section 4, lines 354 – 356 and lines 385 – 390.

3. Why is the VLF(very low frequency) in the frequency domain features not included in the analysis? It is important HRV feature that indicates sympathetic activity.

We agree with the reviewer that the power in the very-low-frequency band [0.003 – 0.04] Hz indicates sympathetic activity and is more strongly associated with mortality than other frequency domain features. However, it requires a recording period of at least 5 minutes to have several periods of variations, and it is best monitored over 24 hours. This feature can thus be calculated in the short-term HRV analysis (5min 30s) but not in the ultra-short-term HRV analysis (1min 30s). Since the objective of the study is to test the validity of the ultra-short-term HRV analysis by comparing it to the short-term HRV analysis, we decided not to calculate the VLF power in the short and ultra-short-term analysis. This decision was already mentioned in the paper in Section 2.2.5 lines 208-209. “The power in the very-low-frequency band was not considered for the analysis of ultra-short duration recordings [21].”

The sentence has been modified to include the reason for excluding this feature in the new version of the paper in Section 2.2.5, lines 209-213, as follows: “The calculation of the power in the very-low-frequency band requires a recording period of at least 5 minutes [21], which is applicable to short-duration recordings but not to ultra-short duration recordings. Therefore, it was not calculated among the frequency domain features.”

Round 2

Reviewer 1 Report

Authors have replied to my queries sufficiently well.

The comment that was perhaps not clear was around projecting the results obtained from a healthy cohort into patients with illness that could affect/vary HR and HRV in the short term and could give false prediction. However since that this is a preliminary study I believe that publication can be granted.

Author Response

We thank the reviewer for their positive comments and for granting permission to publish the paper. We would like to inform you that, in accordance with the suggestion of the second reviewer, we have added a comparison between the ultra-short-term and short-term HRV features computed from the PPG waveforms recorded at a sampling rate of 25 Hz in the results in Section 3.2 lines 324 – 334. The discussion has been updated accordingly.

Reviewer 2 Report

Thanks to the author's sincere reply. My additional concerns are as follows:

1. Please explain in more detail in the paper as mush as you explained to me about overcoming limitations by repeat measurements. 

2. According to the authors' responses, there are three factors that cause to the difference in HRV features; 1) waveform (ECG vs PPG), 2) sampling rate (256fps vs 25fps), 3) recording time (330s vs 90s). The main purpose of the paper is validity of ultra-short-term HRV analysis. It is necessary to compare only the differences by the recording time excluding the effect of other factors (i.e., waveform and sampling rate). I recommend adding the results of comparing the short-term HRV features (330s) with ultra-short-term one (90s) from PPG waveform recorded at a sampling rate of 25fps. 

Author Response

Thanks to the author's sincere reply. My additional concerns are as follows:

 We thank the reviewer for their second round of review of our paper and for their suggestions to improve the quality of the paper. Our responses are presented below.

  1. Please explain in more detail in the paper as mush as you explained to me about overcoming limitations by repeat measurements. 

 As suggested by the reviewer, additional details have been added to the paper. First, we added details on the number of recordings in Section 2.1 “Materials and Methods / Data Collection” (lines 102 – 105). In this section, we found it inappropriate to mention the effect of the multiple recordings per night. Therefore, this point was addressed in Section 2.3 “Materials and Methods / Statistical analysis” (lines 245 – 249) as an introduction to the different statistical methods applied.

  1. According to the authors' responses, there are three factors that cause to the difference in HRV features; 1) waveform (ECG vs PPG), 2) sampling rate (256fps vs 25fps), 3) recording time (330s vs 90s). The main purpose of the paper is validity of ultra-short-term HRV analysis. It is necessary to compare only the differences by the recording time excluding the effect of other factors (i.e., waveform and sampling rate). I recommend adding the results of comparing the short-term HRV features (330s) with ultra-short-term one (90s) from PPG waveform recorded at a sampling rate of 25fps. 

 As the reviewer recommended, we performed the comparison between the ultra-short-term (1 min 30 s) and short-term (5 min 30 s) HRV features computed from the PPG recordings at the same sampling rate. The results of the comparison at the 25 Hz sampling rate are presented in Section 3.2 “Results / Effect of the duration of the recordings” (lines 324 – 334). The detailed results are added as a supplementary file in Table S2-B. The discussion has been updated based on the new analysis performed.